# Mortality from Alcoholic Cardiomyopathy: Exploring the Gap between Estimated and Civil Registry Data

**DOI:** 10.3390/jcm8081137

**Published:** 2019-07-31

**Authors:** Jakob Manthey, Jürgen Rehm

**Affiliations:** 1Institute of Clinical Psychology and Psychotherapy, TU Dresden, Chemnitzer Str. 46, 01187 Dresden, Germany; 2Center for Interdisciplinary Addiction Research (ZIS), Department of Psychiatry and Psychotherapy, University Medical Center Hamburg-Eppendorf (UKE), Martinistraße 52, 20246 Hamburg, Germany; 3Institute for Mental Health Policy Research, CAMH, 33 Russell Street, Toronto, ON M5S 2S1, Canada; 4Dalla Lana School of Public Health, University of Toronto, 155 College Street, 6th Floor, Toronto, ON M5T 3M7, Canada; 5Campbell Family Mental Health Research Institute, CAMH, 250 College Street, Toronto, ON M5T 1R8, Canada; 6Institute of Medical Science (IMS), University of Toronto, Medical Sciences Building, 1 King’s College Circle, Room 2374, Toronto, ON M5S 1A8, Canada; 7Department of Psychiatry, University of Toronto, 250 College Street, 8th Floor, Toronto, ON M5T 1R8, Canada; 8WHO Collaborating Centre for Mental Health and Addiction, 33 Russell Street, Toronto, ON M5S 2S1, Canada; 9Department of International Health Projects, Institute for Leadership and Health Management, I.M. Sechenov First Moscow State Medical University, Trubetskaya str., 8, b. 2, Moscow 119992, Russia

**Keywords:** alcoholic cardiomyopathy, cardiovascular diseases, GBD study, vital statistics, garbage code, cause of death, alcohol per capita consumption, mortality, heart failure

## Abstract

Background: Based on civil registries, 26,000 people died from alcoholic cardiomyopathy (ACM) in 2015 globally. In the Global Burden of Disease (GBD) 2017 study, garbage coded deaths were redistributed to ACM, resulting in substantially higher ACM mortality estimates (96,669 deaths, 95% confidence interval: 82,812–97,507). We aimed to explore the gap between civil registry and GBD mortality data, accounting for alcohol exposure as a cause of ACM. Methods: ACM mortality rates were obtained from civil registries and GBD for *n* = 77 countries. The relationship between registered and estimated mortality rates was assessed by sex and age groups, using Pearson correlation coefficients, in addition to comparing mortality rates with population alcohol exposure—the underlying cause of ACM. Results: Among people aged 65 years or older, civil registry mortality rates of ACM decreased markedly whereas GBD mortality rates increased. The widening gap of registered and estimated mortality rates in the elderly is reflected in a decrease of correlations. The age distribution of alcohol exposure is more consistent with the distribution of civil registry rather than GBD mortality rates. Conclusions: Among older adults, GBD mortality estimates of ACM seem implausible and are inconsistent with alcohol exposure. The garbage code redistribution algorithm should include alcohol exposure for ACM and other alcohol-attributable diseases.

## 1. Introduction

Heavy alcohol use is a major contributor to cardiovascular diseases (CVD) [1,2]. Among other conditions, high levels of alcohol use can lead to alcoholic cardiomyopathy (ACM), which is characterized by a dilation and impairment of the left ventricle [3]. While this condition is the result of the toxic effects of alcohol, the symptoms and clinical presentation are not unique to ACM but largely resemble other dilated cardiomyopathies [4]. As with other cardiomyopathies, ACM is associated with systolic dysfunction and a considerable risk factor for myocardial infarctions and sudden death [4,5]. In the past few years, interest on ACM grew as reflected in the publication of several reviews describing clinical presentations [5], pathophysiological mechanisms [6], as well as the causal relationship between heavy alcohol consumption and incidence of ACM [7]. 

For most alcohol-attributable conditions, mortality is estimated via alcohol-attributable fractions (AAFs) based on alcohol exposure and risk relations. For ACM, however, this method cannot be applied to due to a lack of data quantifying the risk relations [7]. Hence, an alternative method was proposed [8], based on cause of death data reported by countries with civil registration of vital statistics [9]; these vital statistics were available for 91 countries and formed the basis to model ACM as cause of death for all countries and globally [10]. According to these estimates, there were 25,997 global deaths from ACM in 2015. Applying a different methodology, the Global Burden of Disease (GBD) 2017 study resulted in a more than three-fold mortality figure (90,669 deaths in 2015, 95% confidence interval: 82,812–97,507) [11]. Higher GBD estimates were the result of redistributing so called garbage coded deaths to well-defined cause of death codes, including ACM and other CVD [12]. The term garbage code has first been coined in 1996 and refers to all codes, which are not useful for public health analyses [13,14]. More precisely, the term encompasses all codes that are not recognized by the Tenth revision of the International Classification of Diseases (ICD-10, [15]) in describing the actual underlying cause of death. Instead, garbage codes may indicate a symptom (e.g., pain) or an intermediate cause of death, such as heart failure [16].

The outlined gap in ACM mortality estimates warrants further attention given the large difference and because GBD mortality estimates are susceptible to the methods employed in redistributing garbage coded deaths, as previously demonstrated for ischemic heart disease [17]. In this contribution, we sought to further explore the gap between registered and estimated ACM deaths. For this purpose, we used mortality data from civil registries, which serve as one main input for the GBD study, and compare these data with mortality estimates as published in the last GBD update [12]. We aimed to examine the association of registered and estimated ACM mortality rates in countries with available civil registry data. Further, we compared ACM mortality distributions to the distribution of alcohol exposure—its underlying cause by definition. In sensitivity analyses, we examined associations of ACM and garbage coded deaths, which constitute the base for redistributing garbage codes to well-defined causes of death in the GBD study [12]. 

## 2. Experimental Section

### 2.1. Description of Data Sources and Disease Definitions

We obtained adult (15 years or older) mortality data from two sources: (1) civil registration data from the World Health Organization (WHO) mortality database [18], and (2) estimated mortality data from the GBD 2017 study [11]. 

From the WHO mortality database, we retained all country-years with any four-digit ICD-10 code and available data on sex and age. After matching the ICD-10 codes [15] with the disease categories as defined in the GBD 2017 study, we obtained country-, year-, sex-, and age-specific mortality data for the following categories: CVD, cardiomyopathy and myocarditis (hereafter referred to as ‘all cardiomyopathies’), ACM, as well as CVD and heart failure garbage codes (for the definition of each category, see Appendix B and [19]). We calculated the death counts for a new cause of death category ‘all CVD’ from the sum of CVD and CVD garbage code cause of death definition. For country-years with mortality data available only for one sex, we assumed 0 deaths for the other sex. From the GBD database, we obtained the estimated mortality data for the same disease categories (except for garbage codes) [11].

The two mortality data sets were then matched on the above given disease categories and all country-years with available mortality data on ACM and heart failure garbage codes from civil registries were retained, resulting in a total number of 77 countries (with 823 country-years). In order to calculate mortality rates (i.e., deaths per 100,000 adults) we combined the mortality data with population estimates from the UN Population Division [20]. Alcohol exposure was defined as intake of pure alcohol per adult (in liters per year, alcohol *per capita* consumption). Alcohol exposure data were obtained from a recent modeling study using WHO sources and forecasting techniques [21]. Alcohol exposure data was not available for all 5-year age bands, which required the aggregation of mortality data into the available age groups. In addition to same-year, 5-year and 10-year lagged alcohol exposure estimates were calculated to account for the estimated lag time between exposure and disease incidence [7]. These lag times are in line with findings of a recent review and clinical guidelines [7,22].

### 2.2. Descriptive Analyses

For descriptive analyses, we only used data from the most recent available year per country (*n* = 77 data points by sex and age group). For both registered and estimated deaths, we calculated death counts and age-standardized mortality rates per 100,000 adults. To illustrate the gap between registered and estimated deaths, we calculated the ratio of estimated to registered deaths and mortality rates (the larger the ratio, the larger the gap). To examine the association of registered and estimated ACM deaths, Pearson correlations were computed for all adults and by sex and age. 

### 2.3. Sensitivity Analyses

In sensitivity analyses, we aimed to test whether ACM deaths and garbage coded deaths were negatively associated. In the GBD study, a negative association between garbage coded deaths and target diseases is the requirement for the redistribution models [12,13,23]. In brief, these models assume that in jurisdictions with accurate coding practice, a low proportion of deaths are assigned to garbage codes and all other diseases are accurately coded. Consequently, a negative association indicates that the fewer deaths being assigned to garbage codes, the more deaths are being accurately coded. In the GBD study, a negative association between garbage coded deaths and a given cause of death is used as indicator for the redistribution of garbage coded deaths, while positive or non-significant associations indicate that garbage coded deaths may not be redistributed to the given disease.

In this contribution, we performed similar regression models using proportion of ACM deaths among all CVD deaths as target disease and proportion of heart failure deaths among all CVD deaths as garbage codes. We selected heart failure deaths as they were cited as source for redistributing deaths to ACM in the GBD 2017 study [12]. Furthermore, in previous studies the redistribution of heart failure deaths has resulted in an increase of deaths attributable to cardiomyopathy [16,23], which should theoretically increase the number of ACM deaths, as well. 

Poisson regression models examined the relationship of heart failure garbage coded deaths (independent variable) and ACM deaths (dependent variable), allowing for random intercepts in each country. As opposed to GBD redistribution models, we included alcohol *per capita* consumption as additional covariate, which was identified as main driver for estimating ACM mortality [10]. Further, we allowed for nonlinear associations by including polynomials of the independent variable. For more details on the sensitivity analyses, see Appendix B.

All analyses were conducted with *R* version 3.5.1 [24]. 

## 3. Results

A descriptive summary of the mortality data compiled for this study can be found in Table 1. 

### 3.1. Epidemiology of Registered and Estimated ACM Mortality

Using data from the most recent available year for each country, data from *n* = 77 countries could be obtained, representing 1.54 billion adults, mainly living in the Americas and WHO European Region (for a summary on included countries key data, see Appendix A). Since 1990, 63,016 ACM deaths (females: 8816; males: 54,200) were recorded in civil registries. For the same set of countries during the same period, the GBD study estimated the ACM death count at 370,675 (females: 66,080; males: 304,595). Thus, for each registered ACM death, nearly 5 additional deaths have been estimated in the GBD study for the included countries (female ratio: 7.5; male ratio: 5.6). For all cardiomyopathies, the gap between estimated to registered deaths is similar to ACM (female ratio: 8.1; male ratio: 6.3), but it is considerably lower for deaths from all CVD (female ratio: 1.8; male ratio: 1.6).

While the variation by sex was relatively low, there were substantial differences in the distribution of estimated and registered deaths across all ages by cause of death definition. In Figure 1, the registered and estimated mortality rates are presented by sex and across all 5-year age groups (see Appendix B for mortality rates by cause, sex, and age). The age distribution of registered and estimated deaths was largely parallel for CVD, with largely constant ratios of estimated to registered mortality rates across all age groups (1.5–2.0). For all cardiomyopathies, distribution of registered and estimated mortality rates were similar, with exponential increases in older age groups. The ratios of estimated and registered mortality rates for all cardiomyopathies were closer among 15 to 59 year olds (2.4–4.7) and increased in older ages (75 years or older: 9.1–11.4). 

In contrast, the age distribution of registered and estimated mortality rates of ACM diverged substantially. While registered mortality rates largely resemble a normal distribution peaking at ages 60–64 (0.7 deaths per 100,000 adults) and decreasing thereafter, the estimated ACM mortality rates were left-skewed and peaked in the oldest age group (4.7 deaths per 100,000 people aged 85+ years). Consequently, the gap between estimated and registered mortality rates widened with increasing age, with lowest ratios among 25 to 64 year olds (3.2–5.0) and highest ratios in older ages (75 years and older: 13.1–33.4).

The diverging age pattern in ACM mortality figures can also be observed in Figure 2, where registered and estimated mortality rates are presented together with alcohol exposure estimates, for available age groups. The plot suggests that the age distribution of alcohol exposure was more congruent with registered rather than estimated ACM mortality rates. Among older age groups, both registered mortality rates for ACM and alcohol exposure decreased, while the estimated mortality rates increased. Similar patterns can be observed for both same-year, 5-year, and 10-year lagged alcohol exposure.

The age-dependent association of registered and estimated ACM deaths is also presented in Table 2. The correlation of registered and estimated ACM mortality rates was high among all adults for both women and men. Among men, high correlations (>0.55) can be observed for all ages between 20 and 79 years. In older age groups, the correlations were below 0.20. Among women, registered and estimated ACM mortality rates were not associated in the youngest (25–29 years) and oldest (75 and older) age groups, but were associated in the age groups in between. Furthermore, and among both sexes, the association was most pronounced (i.e., >0.7) up to 64-year old people and decreases thereafter.

### 3.2. Sensitivity Analyses

Results of sensitivity analyses are illustrated in Figure 3 (for model results, see Appendix B). Between proportion of ACM deaths and proportion of heart failure deaths among all CVD deaths, there was a non-monotonous negative association. Only for very low proportions of heart failure deaths among all CVD deaths (below 5%) could a decrease in the proportion of ACM deaths among all CVD deaths be observed.

## 4. Discussion

### 4.1. Summary of the Findings

This study compared mortality data from civil registries with estimates from the GBD 2017 study, using data from *n* = 77 countries.

The GBD mortality estimates of ACM seem implausible for the elderly population. Among people aged 65 years or older, the registered ACM mortality rates follow a decrease in alcohol exposure, which is the core determinant of ACM. A similar age distribution of ACM has also been identified in a recent study examining hospitalization data of the United States of America [25]. In contrast, the estimated ACM mortality rates continue to increase with an ageing population. Given a decrease in alcohol exposure among the elderly, an increase of ACM mortality in people aged 65 years or older is unlikely and may result in overestimating the total number of ACM deaths in the GBD study.

### 4.2. Improving ACM Mortality Estimates

For other alcohol-attributable diseases, mortality is estimated using alcohol-attributable fractions, which are calculated from exposure and risk functions (for global mortality estimates, see reference [26]; for a summary of risk functions, see reference [27]). As alcohol-attributable fractions are stratified by age groups, the decline in exposure in the elderly population can be reflected in lower alcohol-attributable fractions in these age groups. For ACM, mortality in the GBD study is estimated by redistributing garbage coded deaths, yet without accounting for age variations in alcohol exposure [12,23]. 

In our view, ACM mortality estimates should be made consistent with alcohol exposure, because this is its core determinant. Further, reductions of alcohol use have been associated with improvements of the clinical course of ACM [28], including mortality risks [29]. In order to align alcohol exposure and ACM mortality estimates in the GBD study, alcohol exposure data should be included in models estimating redistributing proportions for ACM. As indicated in this study, a 5-year or 10-year lag of alcohol exposure may prove useful in redistribution models, which likely represents the period of heavy alcohol intake required to develop ACM [7,22]. While a 5-year period of heavy chronic drinking has been used as lower bound to develop ACM [30,31], up to 25 years of heavy chronic drinking among affected patients have been reported in a number of clinical studies [30,32,33]. However, such long lag times may capture treatment onset rather than disease incidence, similar to the delay between onset and treatment of alcohol use disorders [34]. Thus, in the absence of data from population studies, we proposed to use a 10-year lag to model disease onset until further confirmative data is available. Among 15–24 year olds, a 10-year lag would result in 0 ACM deaths, which is largely in line with the registered deaths and is coherent with alcohol-attributable mortality estimation for cancer [26].

### 4.3. The Impact of Garbage Code Redistribution for ACM Mortality Estimates

In the GBD study, garbage coded deaths are redistributed to other well-defined diseases, including ACM. Since GBD 2013, redistribution proportions are estimated to redistribute garbage coded deaths to selected target diseases (the method proposed in reference [23]; for details of application see Appendix 1 of reference [12]).

Unfortunately, cause-specific results of the redistribution models are not available, thus, it remains unknown how many of the estimated ACM deaths have been redistributed from which garbage code. However, heart failure deaths have been cited as one out of three garbage codes which were redistributed to ACM in the GBD study [12]. Further, heart failure deaths account for a substantial share of CVD deaths in civil registry data and several studies have proposed methods to redistribute heart failure deaths [13,16,23,35]. In brief, heart failure describes an impaired functioning of the heart muscle and is an intermediate state between death and the actual underlying cause, which can be CVDs (e.g., cardiomyopathy, ischemic heart disease) but also other non-communicable diseases such as chronic respiratory diseases, diabetes, or cirrhosis [12,35]. 

Results from our sensitivity analyses suggest that heart failure deaths may not be redistributed to ACM in the majority of countries included in this study. This is in line with previous studies showing only marginal—if any—increases of mortality from all cardiomyopathies after redistributing heart failure garbage codes [16,23]. In GBD, senility and atherosclerosis have been referred to as other garbage codes, which were redistributed to CVD, including ACM [12]. More details on the misclassification of cause of deaths codes should be provided in the GBD study to improve clinical care and cause of death coding practice. For ACM, this is particularly important as five out of six deaths may not be recognized.

### 4.4. Clinical Relevance

There is a large gap between registered and estimated deaths due to ACM but also due to all other cardiomyopathies. Primarily, a large number of deaths assigned with garbage codes may be the result of inaccurate cause of death coding. However, this gap could also be an indicator for suboptimal clinical care (detection, treatment) during a lifetime. To diagnose ACM, clinicians need to identify a dilated heart muscle, rule out other potential causes, and conduct an extensive assessment of the patients’ alcohol use [3,22]. However, the substantial stigma around alcohol dependence [36] may deter clinicians from such conversations with their patients, as reported in primary care [37]. Clinicians may further be discouraged to have discussions around alcohol with their patients because of several uncertainties with this topic, e.g., the differential impact on diseases—especially for cardiovascular diseases, for which both beneficial and detrimental effects have been observed (for an overview of risk functions, see reference [27]), the fact that there is no recognized “safe” level of alcohol consumption (for a recent discussion, see references [38,39,40]), or the lack of an international consensus in defining risky drinking [41].

However, even if all ACM cases were accurately identified in clinical practice (i.e., no garbage codes), the contribution of alcohol may still be underestimated if based on death certificates only [42]. In summary, vital statistics may be the best available data source to estimate ACM mortality to date. Yet, prospective studies on the relationship between alcohol intake and incidence of cardiomyopathy are required to yield more accurate mortality estimates.

### 4.5. Strengths and Limitations

This study used mortality data from 77 countries, representing the majority of high-income countries with highly accurate data from civil registries [9]. However, countries without such data are not represented in this study, most notably African and Asian countries. Thus, the presented results may not apply to non-Caucasian populations from low- and middle-income countries. Further, we restricted the sensitivity analyses to heart failure, which was cited as major source for garbage code redistribution for ACM. As ACM-specific results from the algorithm that redistributes garbage codes in the GBD 2017 study are not available, the impact of redistributing other garbage codes to ACM mortality could not be determined in this study. As the alcohol exposure data used in the GBD study are not available to the authors, we used data from a recent modeling study based on WHO collection and estimation of *per capita* consumption per country, which is considered the most valid estimate of overall alcohol exposure [43], and was validated by country representatives [44]. This has limited the breakup of mortality data to age groups, for which alcohol exposure data is available. 

## 5. Conclusions

GBD mortality estimates of ACM are implausible for adults aged 65 years or older, as they are incongruent with civil registry and alcohol exposure data in this age group. In order to produce more consistent ACM mortality estimates, the redistribution algorithms in the GBD study should be aligned with alcohol exposure data.

## Figures and Tables

**Figure 1 jcm-08-01137-f001:**
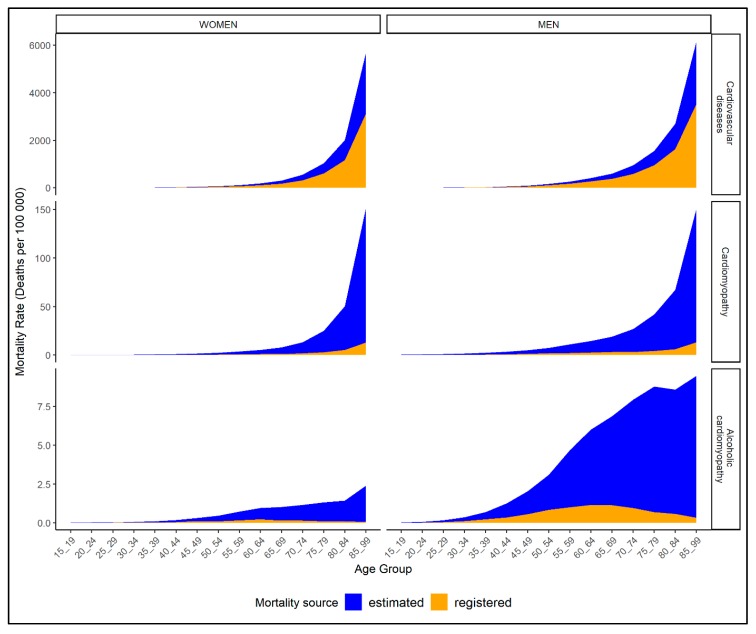
Mortality rates of registered (orange) and estimated (blue) deaths over the life span by cause of death definition (rows) and sex (columns); based on most recent available mortality data from *n* = 77 countries.

**Figure 2 jcm-08-01137-f002:**
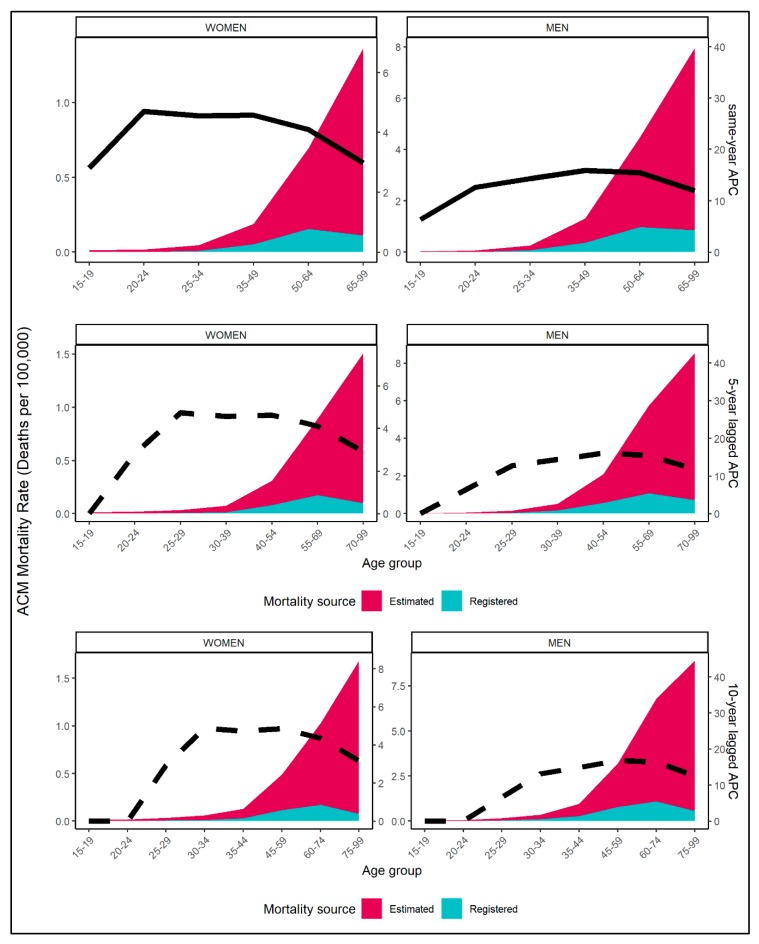
Estimated (red) and registered (blue) ACM mortality rates per 100,000 and alcohol per capita consumption (APC) over selected age groups and by sex (column) for most recent available data of *n* = 77 countries; solid line denotes same-year APC (first row) and dashed line denotes 5-year (second row) and 10-year lagged APC (third row).

**Figure 3 jcm-08-01137-f003:**
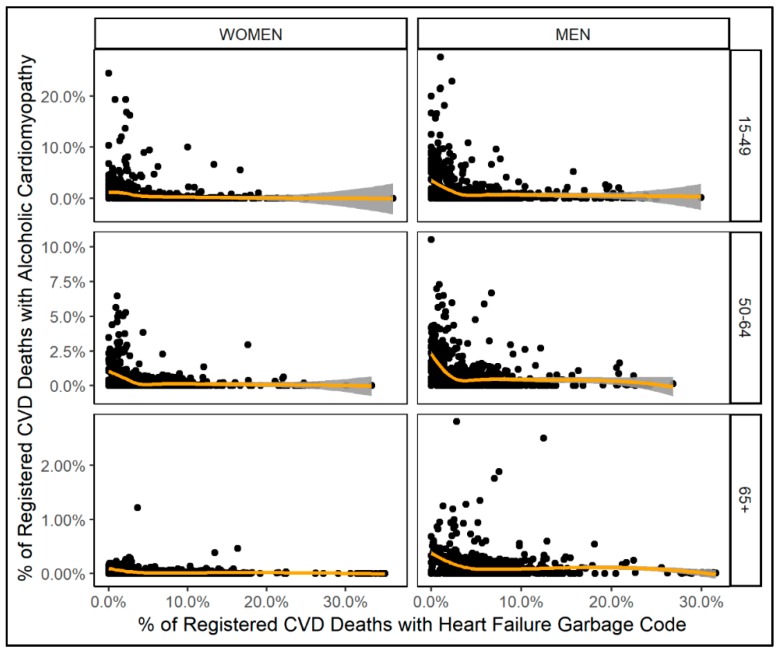
Scatter plots of proportion of heart failure garbage code deaths and proportion of ACM deaths among all CVD deaths by sex (column) and age group (row); orange line denotes the smoothing function of fitted proportion of ACM deaths among all CVD deaths obtained from multi-level models.

**Table 1 jcm-08-01137-t001:** Registered and estimated deaths by disease category and sex.

GBD Disease Definition	Absolute Number of Deaths			Age-Standardized Mortality Rate ^1^		
	Women	Men	Both sexes	Women	Men	Both sexes
**Cardiovascular diseases**						
Registered ^2^	1,396,965	1,469,004	2,865,969	89.1	148.3	116.1
Estimated ^2^	2,459,266	2,386,248	4,845,514	154.9	239.3	193.6
Ratio ^3^	1.8	1.6	1.7	1.7	1.6	1.7
**Cardiomyopathy and myocarditis**						
Registered **^2^**	8,240	11,938	20,178	0.6	1.4	1.0
Estimated ^2^	66,730	75,616	142,346	4.4	8.0	6.1
Ratio ^3^	8.1	6.3	7.1	6.8	5.9	6.2
**Alcoholic Cardiomyopathy**						
Registered ^2^	538	3,345	3,883	0.1	0.4	0.2
Estimated ^2^	3,589	18,894	22,483	0.3	2.1	1.2
Ratio ^3^	6.7	5.6	5.8	5.8	5.4	5.3
**Cardiovascular garbage codes**						
Registered ^2^	963,461	780,529	1,743,990	58.4	77.1	67.2
Estimated ^2^	/	/	/	/	/	/
Ratio ^3^	/	/	/	/	/	/
**Heart failure garbage codes**						
Registered ^2^	283,222	209,033	492,255	15.8	20.0	17.8
Estimated ^2^	/	/	/	/	/	/
Ratio ^3^	/	/	/	/	/	/

Note: Based on *n* = 77 countries, see Appendix A for a detailed list of the included countries. ^1^ Age-standardized number of deaths per 100,000 adults. ^2^ All deaths obtained from in civil registries [18] or as estimated in the GBD 2017 study [11]. ^3^ Ratio of estimated to registered deaths.

**Table 2 jcm-08-01137-t002:** Correlation of registered and estimated ACM crude mortality rates (deaths per 100,000 people) for all adults and by sex and age groups.

	Women	Men
**All Adults**	0.796 (0.697 to 0.866) **	0.917 (0.872 to 0.946) **
**By age group**		
15–19	NA	NA
20–24	NA	0.968 (0.95 to 0.979) **
25–29	0.212 (−0.012 to 0.416)	0.985 (0.977 to 0.991) **
30–34	0.42 (0.217 to 0.589) **	0.988 (0.981 to 0.992) **
35–39	0.731 (0.607 to 0.821) **	0.956 (0.932 to 0.972) **
40–44	0.779 (0.672 to 0.854) **	0.963 (0.942 to 0.976) **
45–49	0.955 (0.93 to 0.971) **	0.941 (0.908 to 0.962) **
50–54	0.956 (0.932 to 0.972) **	0.764 (0.652 to 0.844) **
55–59	0.815 (0.723 to 0.879) **	0.777 (0.67 to 0.853) **
60–64	0.899 (0.845 to 0.935) **	0.901 (0.848 to 0.936) **
65–69	0.305 (0.087 to 0.495) *	0.545 (0.366 to 0.685) **
70–74	0.513 (0.326 to 0.661) **	0.73 (0.605 to 0.82) **
75–79	0.184 (−0.042 to 0.391)	0.618 (0.458 to 0.74) **
80–84	0.099 (−0.127 to 0.316)	0.158 (−0.069 to 0.369)
85–99	0.025 (−0.2 to 0.247)	0.136 (−0.091 to 0.349)

Note: Based on most recent available data from *n* = 77 countries, see Appendix A for a detailed list of the included countries; * *p* < 0.01; ** *p* < 0.001; NA = correlations could not be calculated due to zero registered deaths.

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
