# Peer review of "Mortality from Alcoholic Cardiomyopathy: Exploring the Gap between Estimated and Civil Registry Data"

_jcm, 2019, doi:10.3390/jcm8081137_

Round 1

Reviewer 1 Report

This is an interesting study exploring the gap in mortality from alcoholic cardiomyopathy between estimated and civil registry. Importantly, the authors focus on the correlation with the underlying etiology. The study is well written and the methods are described in detail.

I only have some minor questions regarding their work:

-While reading the manuscript, I was not sure why the authors have decided to explore same-year and 10-year lagged APC instead of 5-five year lagged APC. Of note, a five year lasting heavy alcoholic consumption is required  for the diagnosis of alcoholic cardiomyopathy? ´

- With respect to their data, I would encourage the authors to emphasize, that their results can only be applied to a caucasian polpulation from industrialized countries.

Author Response

This is an interesting study exploring the gap in mortality from alcoholic cardiomyopathy between estimated and civil registry. Importantly, the authors focus on the correlation with the underlying etiology. The study is well written and the methods are described in detail.

I only have some minor questions regarding their work:

Point 1: While reading the manuscript, I was not sure why the authors have decided to explore same-year and 10-year lagged APC instead of 5-five year lagged APC. Of note, a five year lasting heavy alcoholic consumption is required  for the diagnosis of alcoholic cardiomyopathy? ´

Response 1: As suggested, we have added a 5-year lag period to Figure 2, which may represent a lower threshold of heavy chronic drinking required to develop ACM. It is important to note that there is no consensus on the threshold required for establishing a diagnosis. In the statement of the American Heart Association on diagnostic strategies for dilated cardiomyopathies (Reference 30 in the manuscript), it is stated that “Alcoholic cardiomyopathy most commonly occurs in men 30 to 55 years of age who have been heavy consumers of alcohol for >10 years” and “alcoholic patients consuming alcohol for >5 years are at risk for the development of alcoholic cardiomyopathy”. In fact, some clinical studies have reported substantially longer periods (20-25 years) but this might not represent the period between exposure and onset but rather between exposure and treatment. We have altered the discussion section accordingly to account for these studies and to make more clear why we propose to use a 10-year lag period.

Point 2:  With respect to their data, I would encourage the authors to emphasize, that their results can only be applied to a caucasian polpulation from industrialized countries.

Response 1: We agree and have added the following sentence in the limitation section: “Thus, the presented results may not apply to non-Caucasian populations from low- and middle-income countries.”

Reviewer 2 Report

Manthey and Rehm present an interesting study aiming to assess differences between estimated and recorded deaths for alcoholic cardiomyopathy. 

Alcohol use disorders/alcohol abuse represents the first cause of dilated cardiomyopathy after ischemic heart disease. Moreover, alcohol significantly contributes to heart disease acting as an additional mechanism to other causes (i.e. hypertension, etc.). Moreover, the real prevalence of alcohol abuse is underestimated due to patients' underreporting and to physicians' underevaluating attitide. Thus, exploring the gap between the extimated and registered deaths for ACM can help to improve clinical practice.

The paper is interesting and very well written. I have few comments (minor):

1. The title of the first paragraph (i.e. Introduction) is lacking.

2. Introduction section: line 66. I would suggest to better define "garbage codes" to readers.

3. Discussion section: lines 216-217: the words "has also been" have been repeated two times.

4. Discussion section: In addition to discussing the problem of quantifying alcohol consumption, the lack of a linear response to alcohol's toxic effect and the lack of a recognized treshold for "safe" alcohol consumption for ACM could be put in discussion.

5. Discussion section, para 4.4, lines 276: "prospective studies". Given the difficulty in quantifying alcohol consumption I would suggest to change prospective studies with population studies.

Author Response

Manthey and Rehm present an interesting study aiming to assess differences between estimated and recorded deaths for alcoholic cardiomyopathy.

Alcohol use disorders/alcohol abuse represents the first cause of dilated cardiomyopathy after ischemic heart disease. Moreover, alcohol significantly contributes to heart disease acting as an additional mechanism to other causes (i.e. hypertension, etc.). Moreover, the real prevalence of alcohol abuse is underestimated due to patients' underreporting and to physicians' underevaluating attitide. Thus, exploring the gap between the extimated and registered deaths for ACM can help to improve clinical practice.

The paper is interesting and very well written. I have few comments (minor):

Point 1: The title of the first paragraph (i.e. Introduction) is lacking.

Response 1: We have added the title for the introduction.

Point 2: Introduction section: line 66. I would suggest to better define "garbage codes" to readers.

Response 2: We have improved the definition of garbage codes as follows: “The term garbage code has first been coined in 1996 and refers to all codes, which are not useful for public health analyses [13,14].  More precisely, the term encompasses all codes, that are not recognized by the Tenth revision of the International Classification of Diseases (ICD-10, [15]) in describing the actual underlying cause of death.  Instead, garbage codes may indicate a symptom (e.g. pain) or an intermediate cause of death, such as heart failure [16].”

Point 3: Discussion section: lines 216-217: the words "has also been" have been repeated two times.

Response 3: We have removed the repetition.

Point 4: Discussion section: In addition to discussing the problem of quantifying alcohol consumption, the lack of a linear response to alcohol's toxic effect and the lack of a recognized treshold for "safe" alcohol consumption for ACM could be put in discussion.

Response 4: As suggested, we have extended the discussion with these topics: “Clinicians may further be discouraged to have discussions around alcohol with their patients because of several uncertainties with this topic, e.g., the differential impact on diseases – especially for cardiovascular diseases, for which both beneficial and detrimental effects have been observed (for an overview of risk functions, see [27]), the fact that there is no recognized “safe” level of alcohol consumption (for a recent discussion, see [38-40]), or the lack of an international consensus in defining risky drinking [41].”.

Point 5: Discussion section, para 4.4, lines 276: "prospective studies". Given the difficulty in quantifying alcohol consumption I would suggest to change prospective studies with population studies.

Response 5: In revising the discussion section, we have removed this sentence.

Reviewer 3 Report

This is an interesting work about the mortality of alcoholic cardiomyopathy. The author found GBD mortality estimates of ACM seem implausible and are inconsistent with alcohol exposure in elder patients. The garbage code redistribution algorithm should include alcohol exposure for ACM and other alcohol-attributable diseases. In clinical practice, most alcoholic cardiomyopathy may be diagnosed as idiopathic dilated cardiomyopathy. The patients may have a history of alcoholism as drinking the amount of consumed alcohol > 80 g alcohol/day for 20 years. This disease may be not easily diagnosed and need to rule out other possible etiologies of dilated cardiomyopathy. Therefore, prospective studies on the relationship between alcohol intake and incidence of cardiomyopathy are required to yield more accurate mortality estimates and to explore the disease process. However, the work only provides information about the problem of code and did not provide how to improve it.

Author Response

Point 1: This is an interesting work about the mortality of alcoholic cardiomyopathy. The author found GBD mortality estimates of ACM seem implausible and are inconsistent with alcohol exposure in elder patients. The garbage code redistribution algorithm should include alcohol exposure for ACM and other alcohol-attributable diseases. In clinical practice, most alcoholic cardiomyopathy may be diagnosed as idiopathic dilated cardiomyopathy. The patients may have a history of alcoholism as drinking the amount of consumed alcohol > 80 g alcohol/day for 20 years. This disease may be not easily diagnosed and need to rule out other possible etiologies of dilated cardiomyopathy. Therefore, prospective studies on the relationship between alcohol intake and incidence of cardiomyopathy are required to yield more accurate mortality estimates and to explore the disease process. However, the work only provides information about the problem of code and did not provide how to improve it.

Response: Thank you for the positive assessment of our study. We completely agree that prospective, long-term studies are required to improve our understanding of the etiology of ACM. However, we also believe that the contribution of alcohol to the onset of dilated cardiomyopathy can be approached by combining routinely collected cause of death statistics and population alcohol exposure data. We propose to improve the estimation of mortality from ACM by accounting for a 10-year lag of alcohol exposure in the redistribution models. Considering findings of clinical studies, the revised submission now includes an extended discussion on the most appropriate lag time. Further, we have improved the discussion on the clinical implications with regard to the importance of a reliable assessment of alcohol use. However, we believe that recommendations on how to improve clinical practice in this regard would be beyond the scope of this work as these could not be drawn from our findings.

Round 2

Reviewer 3 Report

This is an interesting work about the mortality of alcoholic cardiomyopathy. The work improved much after modification. This work is deserved to publish for alcoholic cardiomyopathy.